# Connecting Joint-Embedding Predictive Architecture with Contrastive Self-supervised Learning

**Shentong Mo**[1]*, **Shengbang Tong**[2]
[1]CMU, [2]NYU

## Abstract

In recent advancements in unsupervised visual representation learning, the Joint-Embedding Predictive Architecture (JEPA) has emerged as a significant method for extracting visual features from unlabeled imagery through an innovative masking strategy. Despite its success, two primary limitations have been identified: the inefficacy of Exponential Moving Average (EMA) from I-JEPA in preventing entire collapse and the inadequacy of I-JEPA prediction in accurately learning the mean of patch representations. Addressing these challenges, this study introduces a novel framework, namely C-JEPA (Contrastive-JEPA), which integrates the Image-based Joint-Embedding Predictive Architecture with the Variance-Invariance-Covariance Regularization (VICReg) strategy. This integration is designed to effectively learn the variance/covariance for preventing entire collapse and ensuring invariance in the mean of augmented views, thereby overcoming the identified limitations. Through empirical and theoretical evaluations, our work demonstrates that C-JEPA significantly enhances the stability and quality of visual representation learning. When pre-trained on the ImageNet-1K dataset, C-JEPA exhibits rapid and improved convergence in both linear probing and fine-tuning performance metrics.

## 1 Introduction

Unsupervised learning of visual representations has recently seen remarkable progress, primarily due to the development of innovative architectures and strategies that exploit unlabeled imagery. Among these advancements, the Joint-Embedding Predictive Architecture (JEPA) [1, 2, 3] has distinguished itself as a powerful approach. I-JEPA [2] leverages a masking strategy to extract visual features, facilitating significant strides in understanding and utilizing unlabeled visual data.

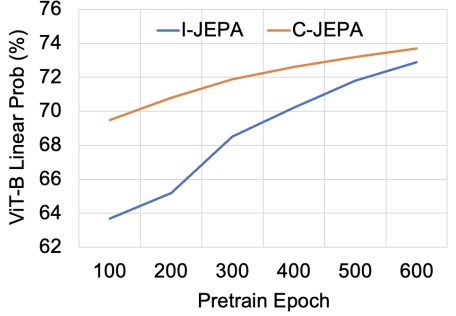

Figure 1: Our C-JEPA achieves faster and better convergence than I-JEPA.

However, despite its successes, certain limitations within the JEPA framework have become apparent, particularly concerning its components I-JEPA Exponential Moving Average (EMA) and I-JEPA prediction capabilities. Specifically, I-JEPA EMA has been found to be inadequate in preventing the issue of entire collapse [4, 5], while the I-JEPA prediction mechanism struggles to accurately learn the mean of patch representations. These challenges not only hinder the performance of JEPA but also limit its applicability in broader contexts.

---

*Corresponding author: shentongmo@gmail.com.

38th Conference on Neural Information Processing Systems (NeurIPS 2024).

To address these limitations, we introduce a novel contrastive self-supervised learning framework based on JEPA, namely C-JEPA, which aims to address the aforementioned challenges by incorporating the principles of Variance-Invariance-Covariance Regularization (VICReg) [6]. VICReg's methodology is adept at learning variance and covariance to avert entire collapse and ensure invariance for the mean of augmented views. By integrating VICReg with the Image-based JEPA, C-JEPA is designed to achieve faster and better convergence, as shown in Figure 1. In this paper, we aim to detail the theoretical underpinnings and empirical validations that substantiate the superiority of C-JEPA over previous self-supervised learning methods.

Our contributions are manifold and significant. Firstly, we identify and articulate the limitations inherent in the I-JEPA framework, specifically its EMA and prediction mechanisms. Secondly, we propose the C-JEPA framework as a novel solution that synergizes JEPA with VICReg to address these limitations effectively. Thirdly, through rigorous empirical and theoretical evaluations, we demonstrate that C-JEPA not only mitigates the issues identified but also achieves superior performance metrics when compared to existing frameworks. Particularly notable is C-JEPA's performance when pre-trained on the ImageNet-1K dataset, where it shows fast and improved convergence in linear probing and fine-tuning scenarios. These results highlight C-JEPA's potential to set a new benchmark in unsupervised visual representation learning, thereby contributing significantly to the field's advancement.

## 2    Related Work

**Self-supervised Learning.** In the self-supervised literature, researchers aim to exploit the internal characteristics of data and leverage pretext tasks to train a model. Recently, an unsupervised framework that learns effective views with data augmentation was proposed by Tian *et al.* [7] to reduce the mutual information between views. CMC [8] introduced a multi-view contrastive learning framework with any number of views to learn view-agnostic representations. Another pretext task of solving jigsaw puzzles was developed in PIRL [9] to improve the semantic quality of learned image representations, achieving better object detection results than supervised pre-training. More recently, Masked image modeling (MIM) has been explored in many previous works [10, 11, 12, 13, 14] to reconstruct the masked image patch given the unmasked counterpart as clues. Some MIM approaches [10, 11, 12, 15, 16] designed customized masking strategies (*i.e.*, random, block-wise) as pre-text tasks during pre-training. For example, block-wise masking was introduced in BEiT [10] to learn transferrable visual representations by recovering discrete tokens of masked image patches. Given features extracted from the 25% unmasked patches, the seminal work, MAE [12] directly reconstructed missing pixels of 75% masked patches. SimMIM [14] randomly masked the input image with a large square patch size (*i.e.*, 32) and used a one-layer prediction layer after the encoder to predict RGB values of raw pixels. Other researchers [15, 16, 17, 18] started to leverage a teacher network like CLIP [19] or adversarial learning to generate the mask and supervision target.

**Contrastive Learning.** In the past years, contrastive learning has shown its effectiveness in self-supervised learning, where various instance-wise contrastive learning frameworks [20, 21, 22, 23, 24, 25, 26, 27, 4] and prototype-level contrastive methods [28, 29, 30, 31] were proposed. The general idea of the instance-wise contrastive learning is to close the distance of the embedding of different views from the same instance while pushing embeddings of views from different instances away. One common way is to use a large batch size to accumulate positive and negative pairs in the same batch. For instance, Chen *et al.* [20] proposed a simple framework with a learnable nonlinear projection head and a large batch size to improve the quality of the pre-trained representations. To make the best use of a large amount of unlabelled data, they present a bigger unsupervised pre-training network and introduce distillation with unlabeled data in SimCLR v2 [21] to improve the performance in downstream tasks. Without involving negative instances, BOYL [22] trains the online network from an augmented view of an image to predict the target network representation of the same image under a different augmented view (positive instance). Another broadly-used approach [23] in the ICL literature is to apply a momentum encoder to update negative instances from a large and consistent dictionary on the fly. The dynamic dictionary was used with a moving-averaged encoder in MoCo series [24, 23] to build a dynamic dictionary to update negative instances in a queue of considerable size. The Variance-Invariance-Covariance (VICReg) [6] regularization strategy has been proposed to address shortcomings in self-supervised learning by enforcing stability through variance and covariance constraints.

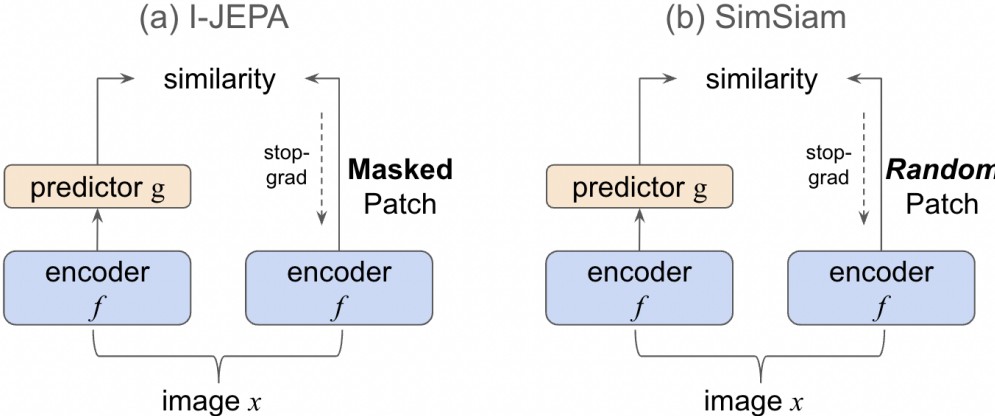

Figure 2: Illustration of I-JEPA (a) and SimSiam (b).

**Joint-Embedding Predictive Architectures.** The Joint-Embedding Predictive Architecture (JEPA) framework has shown considerable promise by enabling the prediction of embeddings from a compatible signal [1]. Building on this, the Image-based Joint-Embedding Predictive Architecture (I-JEPA) introduced by Assran et al. [2] utilizes a context encoder and a target encoder, together with a predictor, to manage representations under a masking regime. This approach focuses on learning embedding spaces directly, contrasting with other methods that operate on pixel or token spaces. While I-JEPA has been successful, it faces challenges such as the inefficacy of Exponential Moving Average (EMA) strategies in preventing model collapse and inaccuracies in predicting the mean of patch representations. These limitations hinder the stability and efficacy of the learned representations, prompting a need for enhanced methodologies. By maintaining sufficient variance in the embeddings and minimizing redundancy among features, VICReg supports robust feature learning. To address the limitations of I-JEPA, our work integrates VICReg into the JEPA framework, thereby aiming to prevent total model collapse and ensure invariance across different views of the same image.

## 3 Methodology

In this section, we present a novel masked modeling framework designed for the joint-embedding predictive architecture to avoid entire collapsing and improve the mean of patch representations. Our key idea is to integrate VICReg into the JEPA framework for alleviating entire model collapse and improving the invariance across different views of the same image. We first provide preliminaries in Section 3.1, then provide the theoretical and empirical connection between I-JEPA and SimSiam in Section 3.2, and finally connect I-JEPA with VICReg in Section 3.3 to show the benefit of reducing entire collapsing and learning the mean of patch representations.

### 3.1 Preliminaries

In this section, we first describe the problem setup and notations, and then revisit I-JEPA [2], SimSiam [4], and VICReg [6] for self-supervised image modeling.

**Problem Setup and Notations.** Given a dataset $\mathcal{X} = \{x_i\}_{i=1}^{N}$ with images $x_i \in \mathbb{R}^{c \times h \times w}$, our goal is to learn a neural network $f_\theta(\cdot)$ to extract unsupervised representations from these visual samples.

**Revisit I-JEPA [2].** Taking the self-supervised modeling as a joint-embedding predictive architecture, I-JEPA [2] utilized $f_\theta(\cdot)$ as a context encoder, a pair of neural network $f'_\theta(\cdot)$ as a target encoder. A predictor $g_\theta(\cdot)$ is applied to predict the target representations from $M$ masked block patches $\mathbf{b}_y(1), ..., \mathbf{b}_y(M)$. For a target block $\mathbf{b}_{y_i}$ corresponding to a target mask $\mathcal{B}_i$, the predictor $g_\theta(\cdot, \cdot)$ takes as input the output of the context encoder $\mathbf{b}_x$ and a mask token for each patch to predict $\{\mathbf{m}_j\}_{j \in \mathcal{B}_i}$, and outputs the patch-level prediction $\{\hat{\mathbf{b}}_{y_j}\}_{j \in \mathcal{B}_i}$, that is, $\{\hat{\mathbf{b}}_{y_j}\}_{j \in \mathcal{B}_i} = g_\theta(\{\mathbf{m}_j\}_{j \in \mathcal{B}_i})$.

The masking objective is optimized by the average $\ell_2$ distance between the predicted patch-level representations $\hat{\mathbf{b}}_{y_j}$ and the target patch-level representation $\mathbf{b}_{y_j}$, which is formulated as:

$$\mathcal{L}_{\text{I-JEPA}} = \frac{1}{|M|} \sum_{i=1}^{M} \sum_{j \in \mathcal{B}_i} ||\hat{\mathbf{b}}_{y_j} - \mathbf{b}_{y_j}||_2^2, \tag{1}$$

where $|M|$ denotes the total number of target blocks $M$, and $\mathcal{B}_i$ is the generated mask corresponding to the $i$-th target block during training.

**Revisit SimSiam [4].** For contrastive self-supervised learning, SimSiam [4] tried to obtain two augmented views $x_i$ and $x_i'$. Then they fed two views into a pair of neural networks $f_\theta(\cdot)$ and $f_\theta'(\cdot)$. To learn the invariance of two different views from the same image, a prediction MLP head, denoted as $g$, transforms the output $\mathbf{z}$ of one view and matches it to representations $\mathbf{p}$ of the other view. The overall objective of SimSiam is to minimize the distance between $\mathbf{z}$ and $\mathbf{p}$ for all random patches $r_j$ in the same image, which is formulated as:

$$\mathcal{L}_{\text{SimSiam}} = \frac{1}{|V|} \sum_{i=1}^{V} \sum_{j \in \mathcal{P}_i} ||\mathbf{z}_{r_j} - \mathbf{p}_{r_j}||_2^2, \tag{2}$$

where $|V|$ denotes the total number of augmented views $V$, and $\mathcal{P}_i$ is the all random patches corresponding to the $i$-th view during training.

**Revisit VICReg [6].** To prevent a collapse in which the encoders produce constant or non-informative vectors in Siamese net architecture, VICReg [6] introduced variance and covariance regularization terms based on the representations space in the invariance terms. Firstly, the variance regularization term $v$ as a hinge function on the standard deviation of the embeddings $\mathbf{z}$ along the batch dimension $n$ to prevent collapse with all the embeddings mapped on the same vector, which is denoted as:

$$v(\mathbf{z}) = \frac{1}{d} \sum_{j=1}^{d} \max(0, \gamma - \sqrt{\text{Var}(\mathbf{z}_j) + \epsilon}), \tag{3}$$

where $\gamma = 1$ denotes a constant value for the standard deviation in their experiments, and $\epsilon$ denotes a small numerical scalar to stabilize training. Secondly, the covariance regularization term $c$ minimizes the sum of the squared off-diagonal coefficients to encourage the off-diagonal coefficients of correlation matrix $C$ along the batch dimension to be close to 0, which is formulated as:

$$c(\mathbf{z}) = \frac{1}{d} \sum_{i \neq j} [\frac{1}{n-1} \sum_{i=1}^{n} (\mathbf{z}_i - \bar{\mathbf{z}})(\mathbf{z}_i - \bar{\mathbf{z}})^T)]_{i,j}^2, \tag{4}$$

where $\bar{\mathbf{z}} = \frac{1}{n} \sum_{i=1}^{n} \mathbf{z}_i$. This term is helpful to prevent different dimensions of the embeddings from encoding trivial information. Finally, the invariance term is defined to minimize the mean-squared Euclidean distance between each pair of vectors $\mathbf{z}$ and $\mathbf{z}_{r_j}$ from different views.

## 3.2 Connecting I-JEPA with SimSiam

In the realm of self-supervised learning, both I-JEPA [2] and SimSiam [4] frameworks aim to extract robust representations from images, yet they approach the problem with distinct architectures and objectives. Here, we draw theoretical and empirical parallels between these methodologies, leveraging their unique approaches to enhance the understanding of joint-embedding architectures.

Regarding theoretical connections, I-JEPA [2] uses a predictive model where the encoder and predictor work together to forecast masked parts of the input, relying on partial views. Conversely, SimSiam [4] operates without explicit masking, utilizing dual augmentations of the same image to enforce consistency between the independently processed views. Both models share the underlying principle of minimizing the distance between certain representations. As shown in Figure 2, I-JEPA focuses on the distance between predicted and actual masked patch representations, whereas SimSiam minimizes the distance between the two augmented views of an image, thereby encouraging consistency across different transformations of the same data.

Meanwhile, empirical studies such as Tian et al. [5] have shown that linear predictors in frameworks like BYOL [22], which are closely related to SimSiam, tend to learn alignment with the correlation

matrix of representations. This suggests that SimSiam's approach without a separate predictor could inherently align with the mean of augmented views, a concept central to I-JEPA's strategy of predicting masked patches. A study [32] in joint-embedding self-supervised learning also indicates that these methods primarily focus on capturing the co-occurrence and distribution of image patches, which aligns with I-JEPA's objective of learning from masked representations.

### 3.3 Connecting I-JEPA with VICReg

VICReg [6] introduces variance and covariance regularization to prevent the collapse of representations in Siamese networks, ensuring that the model learns informative and diverse features across different dimensions. By integrating VICReg into I-JEPA, our objective is to tackle the common challenges of model collapse and enhance the mean representation learning of image patches.

VICReg's variance regularization ensures that all dimensions of the embedding space contain meaningful variance, which is crucial for preventing the model from collapsing to trivial solutions. I-JEPA, which aims to learn diverse patch representations, can benefit from such a mechanism to ensure that each masked patch contributes informatively to the overall representation. By minimizing the off-diagonal elements of the covariance matrix, VICReg encourages the features to be uncorrelated, enhancing the diversity of the learned features. This aspect can be particularly beneficial for I-JEPA, where diverse patch predictions are essential for effective representation learning. Literature on non-contrastive self-supervised learning [33] suggests that implicit variance regularization, as seen in VICReg, can facilitate the learning dynamics in joint-embedding architectures. Such regularization helps maintain a balance between similarity and diversity in learned representations, which is crucial for effective self-supervised learning.

In the following, we consider linear predictor $W_P \in \mathbb{R}^{M \times M}$ in I-JPEA, the masking objective with the average $\ell_2$ distance between the predicted patch-level representations $\mathbf{z}_{y_j}$ and the target patch-level representations $\mathbf{z}_{y_j}^a$ as

$$\mathcal{L} = \frac{1}{|M|} \sum_{i=1}^{M} \sum_{j \in \mathcal{B}_i} ||W_P \mathbf{z}_{y_j} - \text{SG}(\mathbf{z}_{y_j}^a)||_2^2, \tag{5}$$

where $|M|$ denotes the total number of target blocks $M$, and $\mathcal{B}_i$ is the generated mask corresponding to the $i$-th target block during training. By connecting I-JEPA with SimSiam in Section 3.2, we can can diagonalize correlation matrix of $\mathbf{z}_{y_j}$ over $\mathbb{R}$: $C_{\mathbf{z}_{y_j}} = U D_C U^T$, where $U$ is an orthogonal matrix whose columns are the eigenvectors of $C_{\mathbf{z}_{y_j}}$ and $D_C$ is the real-valued diagonal matrix of the eigenvalues $s_k, k \in [1, K]$. Given this eigendecomposition, the predictor is directly set to $W_P = U D_C^\alpha U^T$, where $\alpha$ is a positive constant exponent applied element-wise to $D_C$. The eigenvalues $\lambda_k$ of the predictor matrix $W_P$ are then $\lambda_k = s_k^\alpha$. Assume $\hat{\mathbf{z}}_{y_j}$ the representations expressed in the predictor's eigenbasis, the asymmetric loss $\mathcal{L}$ can be formulated as:

$$\mathcal{L} = \frac{1}{|M|} \sum_{i=1}^{M} \sum_{j \in \mathcal{B}_i} \sum_{k}^{K} ||\lambda_k \hat{\mathbf{z}}_{y_j} - \text{SG}(\hat{\mathbf{z}}_{y_j}^a)||_2^2, \tag{6}$$

Following non-contrastive self-supervised learning [33], we can use NTK [34, 35] to characterize the learning gradient dynamics of neural networks as $\nabla_{\mathbf{z}_{y_j}} \mathcal{L} = (D\mathbf{z}_{y_j} - \mathbf{z}_{y_j}^t)D$, and the representational dynamics of each mode $k$ independently follow gradient of the loss $-\nabla_{\hat{\mathbf{z}}_{y_j}} \mathcal{L}$, and decouple as $K$ independent differential equations:

$$\frac{d\hat{\mathbf{z}}_{y_j,k}}{dt} = -\eta \frac{\partial \mathcal{L}}{\partial \hat{\mathbf{z}}_{y_j,k}}(t) = \eta \lambda_k \left( \hat{\mathbf{z}}_{y_j,k}^a - \lambda_k \hat{\mathbf{z}}_{y_j,k} \right) \tag{7}$$

By taking the expectation over the same masking blocks, we can have the dynamics for each mask patch $y_j$ as

$$\frac{d\hat{\mathbf{z}}_{y_j,k}}{dt} = \eta \lambda_k (1 - \lambda_k) \hat{\mathbf{z}}_{y_j,k} \tag{8}$$

Note that when $\lambda_k < 1$, $d\hat{\mathbf{z}}_{y_j,k}$ has the same sign as $\hat{\mathbf{z}}_{y_j,k}$ and they have opposite sign at $\lambda_k > 1$. These convergent dynamics will push an eigenvalue $\lambda_k$ of one to prevent the collapse of each mode $k$ in representation dynamics. When removing the stop-grad operator in $\mathcal{L}$, we can have

$\frac{d\hat{\mathbf{z}}_{y_j,k}}{dt} = -\eta(1-\lambda_k)^2\hat{\mathbf{z}}_{y_j,k}$, where $d\hat{\mathbf{z}}_{y_j,k}$ and $\hat{\mathbf{z}}_{y_j,k}$ always have opposite signs and the learned representations will become zero with exponentially decaying eigenvalues to collapse. Without the predictor $W_P$, we will have $\frac{d\hat{\mathbf{z}}_{y_j,k}}{dt} = 0$, and the representations will not be updated.

Incorporating VICReg's regularization strategies into I-JEPA could prevent the entire collapsing of the model, especially when learning from a large and diverse dataset. This integration could also improve the granularity and utility of the patch-level representations by ensuring that each dimension of the embedding space remains informative and independent. Overall, the integration of VICReg and SimSiam principles into the I-JEPA framework offers promising avenues for enhancing the robustness and efficacy of self-supervised learning models, particularly in tasks requiring nuanced understanding of complex visual content.

## 4 Experiments

### 4.1 Experimental setup

**Datasets.** Following previous methods [12, 2], we use ImageNet-1K [36] for image classification, MS-COCO [37] for object detection and instance segmentation, and ADE20K [38, 39] for semantic segmentation. We closely follow previous work [40, 41], and adopt the Mask R-CNN [42] as the detector. The ViTs [43] backbone weights are initialized with weights pre-trained on ImageNet-1K using our C-JEPA. Following the settings in [12], we use the Semantic FPN and UPerNet approach [44] based on our ImageNet-1K pre-trained ViTs for evaluation. For a fair comparison, we fine-tune the detector with the same learning rate in [12]. For video object segmentation, we use DAVIS-2017 dataset containing 60 training, 30 validation, and 60 testing videos. For low-level tasks, we follow the previous work [2] and use Clevr/Count and Clevr/Dist on Clevr [45] dataset.

**Evaluation Metrics.** For image classification, we follow previous masked image modeling methods [12, 2] to report the classification accuracy of linear probing and fine-tuning. For object detection and instance segmentation on MS-COCO, we apply AP$^{\text{box}}$ and AP$^{\text{mask}}$ as metrics for the bounding boxes and the instance masks. mIoU results are reported to evaluate semantic segmentation on ADE20K. For video object segmentation on DAVIS-2017, we use Jabri-based $(\mathcal{J}\&\mathcal{F})_m$, $\mathcal{J}_m$, $\mathcal{F}_m$ as metrics to evaluate the quality of frozen representations of image patches by segmenting scenes with the nearest neighbor between consecutive frames. For object counting and depth prediction tasks on Clevr, we use object counting and depth prediction to evaluate the linear probing performance of our model.

**Implementation.** For input images, we resized the resolution to $224 \times 224$, $i.e.$, $H = W = 224$. Following prior work [12, 2], we apply a patch size of 16, $i.e.$, $P = 16$. We use the tiny, small, base, and large models of ViT [43] architecture for experiments. We set the embedding dimension of the predictor to 384, and keep the number of self-attention heads the same as the backbone context-encoder. For the ViT-T/16, ViT-S/16, and ViT-B/16 context-encoder, we set the depth of the predictor as 6. For ViT-L/16 context-encoders, we set the depth of the predictor to 12. Following I-JEPA [2], we use AdamW to optimize the context-encoder and predictor weights. We train our model using the default batch size of 2048, and the learning rate linearly increased from 1e-4 to 1e-3 during the first 15 epochs of pre-training, and decay to 1e-6 following a cosine schedule. The weight decay is linearly increased from 0.04 to 0.4, and the target-encoder weights are initialized the same as the context-encoder weights, and updated via an exponential moving average. We use a momentum value of 0.996, and linearly increase this value to 1.0. For masking, we use the same strategy and settings as I-JEPA [2] for 4 possibly overlapping target block masks.

### 4.2 Experimental comparisons

In this work, we propose a novel and effective framework for connecting Joint-Embedding Predictive Architecture with VICReg in non-contrastive self-supervised learning. In order to demonstrate the effectiveness of the proposed C-JEPA, we comprehensively compare it to previous baselines [41, 10, 12, 46, 47, 2] on non-contrastive self-supervised learning and mask image modeling.

**ImageNet-1K image classification.** For image classification on the ImageNet-1K benchmark, we report the quantitative comparison results on linear evaluation and fine-tuning results in Table 1. We can observe that our C-JEPA achieves the best results in terms of ViT-B/16 compared to previous

Table 1: **Comparison with prior work.** We perform linear evaluation, fine-tuning, COCO detection/segmentation, and ADE20K semantic segmentation on pre-trained ViT-B/16 models. We report linprob, fine-tune, $AP^{box}$, $AP^{mask}$, and mIoU to evaluate the quality of pre-trained representations. The best results are indicated in **bold**.

| Method | Pretrain Epochs | linprob | fine-tune | $AP^{box}$ | $AP^{mask}$ | mIoU |
|---|---|---|---|---|---|---|
| DINO [41] | 1600 | 78.2 | 82.8 | 50.1 | 43.4 | 46.8 |
| BEiT [10] | 800 | 56.7 | 83.4 | 49.8 | 44.4 | 47.1 |
| MAE [12] | 1600 | 68.0 | 83.6 | 50.3 | 44.9 | 48.1 |
| iBOT [46] | 1600 | 79.5 | 84.0 | 51.2 | 44.2 | 50.0 |
| data2vec [47] | 800 | 60.8 | 84.2 | – | – | 48.2 |
| I-JEPA [2] | 600 | 72.9 | 83.5 | 49.9 | 44.5 | 47.6 |
| C-JEPA (ours) | 600 | **73.7** | **84.5** | **50.7** | **45.3** | **48.7** |

Table 2: **Scaling up to Large Models.** We perform linear evaluation, fine-tuning, video object segmentation, and low-level tasks on pre-trained ViT-L/16 models. We report linprob, fine-tune, $(\mathcal{J}\&\mathcal{F})_m$, Clevr/Count and Clevr/Dist metrics to evaluate the quality of pre-trained representations. The best results are indicated in **bold**.

| Method | Pretrain Epochs | linprob | fine-tune | $(\mathcal{J}\&\mathcal{F})_m$ | Clevr/Count | Clevr/Dist |
|---|---|---|---|---|---|---|
| I-JEPA [2] | 600 | 77.5 | 85.3 | 56.6 | 85.6 | 71.2 |
| C-JEPA (ours) | 600 | **78.1** | **86.2** | **58.3** | **86.8** | **71.6** |

masked image modeling approaches. Specifically, the proposed method outperforms the I-JEPA by 0.8 and 1.0 in terms of top-1 accuracy on linear evaluation and fine-tuning settings. Meanwhile, we enjoy the advantage of fewer pre-training epochs compared to other masked image model frameworks.

**COCO object detection and instance segmentation.** For COCO object detection & instance segmentation benchmarks, we also report the quantitative comparison results in Table 1. As can be seen, we achieve significant performance gains of $0.8@AP^{box}$ and $0.8@AP^{mask}$ on COCO object detection and instance segmentation compared to I-JEPA. We also achieve better results than DINO [41] and MAE [12] regarding both settings.

**ADE20K semantic segmentation.** For the ADE20K semantic segmentation, we report the quantitative comparison results in Table 1. As can be seen, the proposed C-JEPA outperforms I-JEPA [2], the current image-based joint-embedding predictive architecture by 1.1@mIoU. Also, we observed that our C-JEPA can achieve better performance than other masked image modeling baselines, including MAE [12] and data2vec [47]. These improvements suggest the importance of leveraging the VICReg's regularization to capture better semantics for dense prediction tasks.

**DAVIS video object segmentation.** In addition, we scale up our experiments from ViT-B to ViT-L and report the results in Table 2. The proposed C-JEPA still achieves better performance than I-JEPA [2] in terms of linear evaluation and fine-tuning settings. For DAVIS video object segmentation, our method also shows significant and consistent gains as shown in Table 2. Compared to I-JEPA, ours achieved the results gains of $1.7@(\mathcal{J}\&\mathcal{F})_m$ on ViT-L/16.

**Clevr object counting and depth prediction.** We further measure the abilities of object-counting and depth prediction on Clevr benchmarks for our pre-trained models. Table 2 shows linear evaluation results of C-JEPA on the Clevr counting and depth benchmarks. Compared to I-JEPA [2], we achieve the results gains of 1.2@Clevr/Count and 0.4@Clevr/Dist using ViT-L/16. These results further indicate the benefit of the proposed method in learning better representations than our vanilla I-JEPA baseline without VICReg regularization.

### 4.3 Experimental analysis

In this section, we performed ablation studies to demonstrate the benefit of the proposed Variance/Covariance and Invariance terms from VICReg. Here, we conducted extensive experiments on ImageNet-1k pre-trained ViT-B/16 to explore the impact of representation mean and collapse, and learned meaningful qualitative patch-level representations.

Table 3: **Ablation studies on component analysis for faster convergence.** We perform ablation studies on Variance/Covariance and Invariance modules in VICReg using ViT-B/16 model. The best results are indicated in **bold**.

| I-JEPA | Variance/Covariance | Invariance | Backbone | Pretrain Epoch | linprob | fine-tune | (J & F)_m |
|---|---|---|---|---|---|---|---|
| ✓ | ✗ | ✗ | ViT-B/16 | 100 | 63.7 | 82.5 | 52.3 |
| ✓(mean) | ✓(collapse) | ✗ | ViT-B/16 | 100 | 68.3 | 83.2 | 54.6 |
| ✓ | ✓ | ✓ | ViT-B/16 | 100 | **69.5** | **83.6** | **55.2** |
| ✓(EMA for collapse) | ✗ | ✓(mean) | ViT-B/16 | 100 | 67.6 | 82.8 | 53.9 |

Table 4: **Ablation studies on component analysis for better convergence.** We perform ablation studies on Variance/Covariance and Invariance modules in VICReg using ViT-B/16 model. The best results are indicated in **bold**.

| I-JEPA | Variance/Covariance | Invariance | Backbone | Pretrain Epoch | linprob | fine-tune | (J & F)_m |
|---|---|---|---|---|---|---|---|
| ✓ | ✗ | ✗ | ViT-B/16 | 600 | 72.9 | 83.9 | 56.2 |
| ✓(mean) | ✓(collapse) | ✗ | ViT-B/16 | 600 | 73.5 | 84.3 | 56.9 |
| ✓ | ✓ | ✓ | ViT-B/16 | 600 | **73.7** | **84.5** | **57.5** |
| ✓(EMA for collapse) | ✗ | ✓(mean) | ViT-B/16 | 600 | 73.2 | 84.2 | 56.6 |

**Variance/Covariance and Invariance in VICReg.** We first conducted an ablation study to determine the efficacy of incorporating Variance and Covariance terms, along with the Invariance term, in preventing model collapse and improving the fidelity of representation means. These experiments are crucial for understanding how each component contributes to the overall stability and effectiveness of the learned embeddings. The results, as presented in Tables 3 and 4, provide clear insights into the performance enhancements driven by these modifications. The inclusion of the Variance and Covariance terms significantly accelerated the training process, as evidenced by the metrics in Table 3. Table 4 highlights improved accuracy and stability in the embeddings, demonstrating the beneficial impact of the Invariance term in aligning the representation means across different mask blocks of the same images.

**Representation Mean and Collapse.** Further analysis focused on how well the C-JEPA framework mitigates the risk of model collapse and accurately learns the mean of patch representations: The integration of VICReg's terms with C-JEPA's architecture addresses the previously noted deficiencies in I-JEPA's EMA component, which was prone to collapsing. The results (referenced in the last rows of Tables 3 and 4) underscore the robustness added by these regularization terms from VICReg.

**Qualitative Attention Visualizations.** To complement our quantitative findings, we also examined qualitative aspects through attention visualization techniques. Figures 3 and **??** showcase the attention maps generated by both the base I-JEPA and the enhanced C-JEPA models. These visualizations illustrate the more focused and contextually relevant attention in C-JEPA, which correlates with the theoretical improvements expected from incorporating VICReg's terms. The attention maps further validate our approach by visually demonstrating the enhanced capability of C-JEPA to maintain stable and meaningful patch-level representations across various image contexts, an improvement over the base I-JEPA model. This qualitative evidence supports the quantitative improvements and highlights the benefits of our approach in capturing invariance.

## 5 Conclusion

In this work, we introduced C-JEPA, a novel enhancement to the Joint-Embedding Predictive Architecture that integrates the robust features of Variance-Invariance-Covariance Regularization (VICReg) to address critical limitations in the existing I-JEPA model. By refining the EMA and prediction components of I-JEPA, C-JEPA successfully prevents entire model collapse and more effectively learns the mean of patch representations, thereby advancing the state of unsupervised visual representation learning. Our theoretical analysis and empirical results have demonstrated the effectiveness of C-JEPA, particularly when pre-trained on the ImageNet-1K dataset. The framework shows marked improvements in convergence rates and performance in both linear probing and fine-tuning scenarios, outperforming existing methods. This underscores C-JEPA's capability to not only address the weaknesses of its predecessor but also to provide substantial improvements in learning quality and stability.

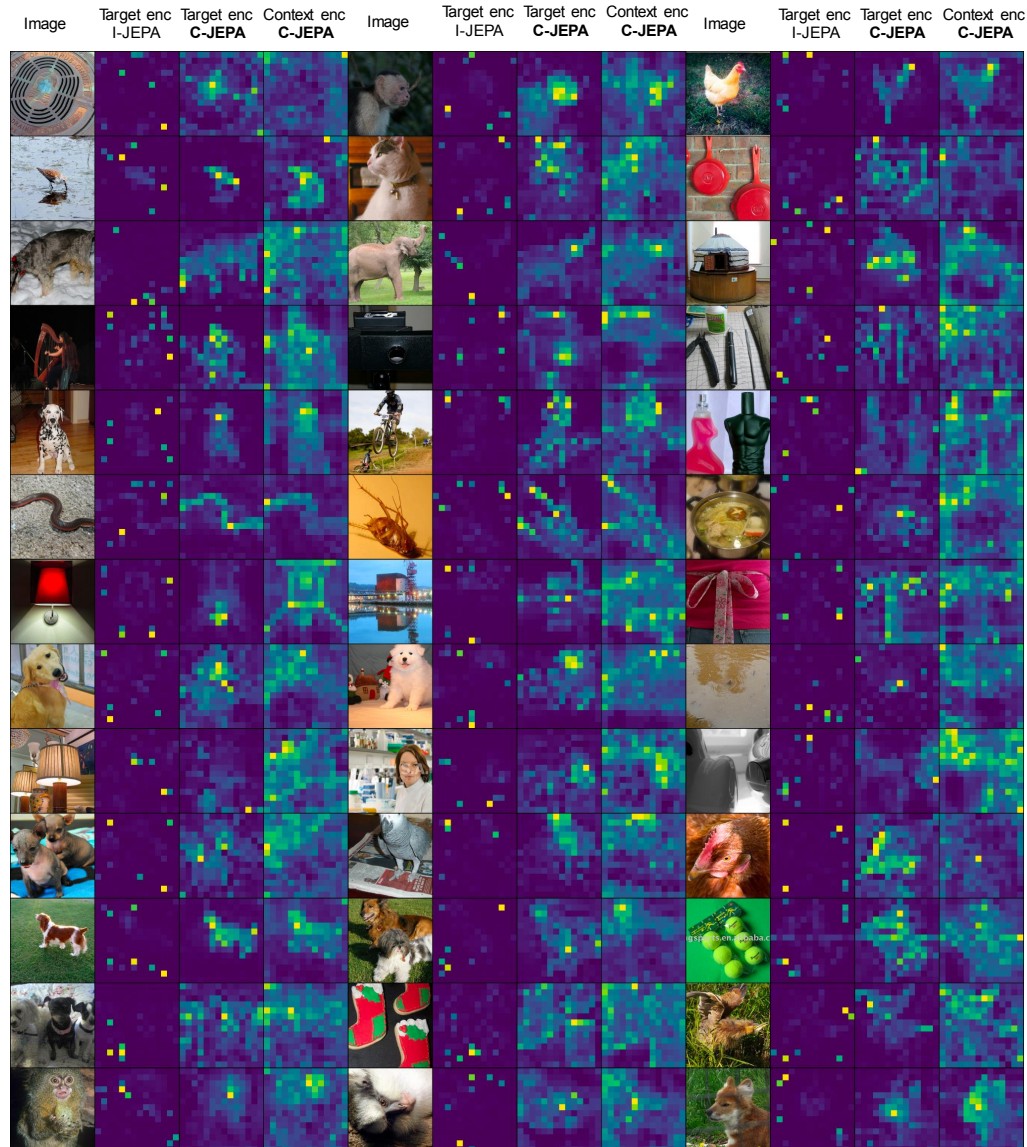

Figure 3: **Qualitative visualization of learned attention maps using ViT-B/16 model.** Columns for each sample denote the original image, attention maps from the target encoder in I-JEPA [2], attention maps from the target encoder in our C-JEPA, and attention maps from the context encoder in our C-JEPA. Our C-JEPA achieves much better attention maps.

## Broader Impact

Our C-JEPA's adoption of VICReg principles enhances its versatility and applicability across various unsupervised learning contexts, making it a valuable tool for tasks requiring robust and reliable visual representations. The implications of this research are significant, offering a pathway for future studies to explore and expand upon the integration of contrastive and joint-embedding techniques. As we move forward, further refinements and explorations into the scalability of C-JEPA in more diverse and challenging datasets will be crucial. Additionally, investigating the adaptability of C-JEPA to different domains and its effectiveness in real-world applications will be essential to fully realize its potential. C-JEPA represents a significant step forward in the field of machine learning, particularly in the unsupervised learning of visual representations. It sets a new benchmark for future research and development in this area, promising enhanced performance and broader applicability in tackling complex visual understanding tasks.

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

# Appendix

In this appendix, we provide the following materials:

- proofs on the loss $\mathcal{L}$ in the predictor eigenspace and representation dynamics under $\mathcal{L}$ Section A,
- derivation of learning dynamics without the stop-grad and predictor in Section B,
- detailed loss functions and algorithm in Section C,
- additional experimental results in small-scale models and VICReg hyperparameters in Section D,
- additional visualizations on the learned attention maps in Section E.

## A  Proofs

### A.1  Loss $\mathcal{L}$ in the predictor eigenspace

We consider linear predictor $W_P \in \mathbb{R}^{M \times M}$ in I-JPEA, the masking objective with the average $\ell_2$ distance between the predicted patch-level representations $\mathbf{z}_{y_j}$ and the target patch-level representations $\mathbf{z}_{y_j}^a$ as

$$\mathcal{L} = \frac{1}{|M|} \sum_{i=1}^{M} \sum_{j \in \mathcal{B}_i} ||W_P \mathbf{z}_{y_j} - \mathrm{SG}(\mathbf{z}_{y_j}^a)||_2^2, \tag{9}$$

Assume $\hat{\mathbf{z}}_{y_j}$ the representations expressed in the predictor's eigenbasis. Since $W_P \in \mathbb{R}^{M \times M}$ is a symmetric matrix with eigendecomposition $W_P = U D_C U^T$, where $U$ is an orthogonal matrix and $UU^T = I$, we can substitute $W_P$ in Eq. 9 as

$$\begin{aligned}
\mathcal{L} &= \frac{1}{|M|} \sum_{i=1}^{M} \sum_{j \in \mathcal{B}_i} ||W_P \mathbf{z}_{y_j} - \mathrm{SG}(\mathbf{z}_{y_j}^a)||_2^2 \\
&= \frac{1}{|M|} \sum_{i=1}^{M} \sum_{j \in \mathcal{B}_i} ||U D_C U^T \mathbf{z}_{y_j} - \mathrm{SG}(UU^T \mathbf{z}_{y_j}^a)||_2^2 \\
&= \frac{1}{|M|} \sum_{i=1}^{M} \sum_{j \in \mathcal{B}_i} ||D_C \mathbf{z}_{y_j} - \mathrm{SG}(\mathbf{z}_{y_j}^a)||_2^2 \\
&= \frac{1}{|M|} \sum_{i=1}^{M} \sum_{j \in \mathcal{B}_i} \sum_{k}^{K} ||\lambda_k \hat{\mathbf{z}}_{y_j} - \mathrm{SG}(\hat{\mathbf{z}}_{y_j}^a)||_2^2
\end{aligned} \tag{10}$$

### A.2  Representational dynamics under $\mathcal{L}$

Following non-contrastive self-supervised learning [33], we can use NTK [34, 35] to characterize the learning gradient dynamics of neural networks as $\nabla_{\mathbf{z}_{y_j}} \mathcal{L} = (D\mathbf{z}_{y_j} - \mathbf{z}_{y_j}^t)D$, and the representational dynamics of each mode $k$ independently follow gradient of the loss $-\nabla_{\hat{\mathbf{z}}_{y_j}} \mathcal{L}$, and decouple as $K$ independent differential equations:

$$\frac{d\hat{\mathbf{z}}_{y_j,k}}{dt} = -\eta \frac{\partial \mathcal{L}}{\partial \hat{\mathbf{z}}_{y_j,k}}(t) = \eta \lambda_k \left( \hat{\mathbf{z}}_{y_j,k}^a - \lambda_k \hat{\mathbf{z}}_{y_j,k} \right) \tag{11}$$

By taking the expectation over the same masking blocks, we can have the dynamics for each mask patch $y_j$ as

$$\frac{d\hat{\mathbf{z}}_{y_j,k}}{dt} = \eta \lambda_k (1 - \lambda_k) \hat{\mathbf{z}}_{y_j,k} \tag{12}$$

Suppose the parameters of the neural network are denoted as $\theta$ with weights $W \in \mathbb{R}^{M \times M}$, we follow Lemma 3 proposed in [33] and have the empirical NTK $\hat{\Theta}(\mathcal{Y}, \mathcal{Y})$ in the orthogonal eigenbasis is equal

to the empirical NTK$\Theta(\mathcal{Y}, \mathcal{Y})$ in the original basis. Each block of the full empirical NTK is denoted as $\hat{\Theta}_t(\boldsymbol{y}_j, \boldsymbol{y}_q) = (\boldsymbol{y}_j^T \boldsymbol{y}_q)I_M$, where $\boldsymbol{y}_j, \boldsymbol{y}_q$ denote the high-dimensional inputs from mask patches $j$ and $q$ in the training samples. $I_M \in \mathbb{R}^{M \times M}$ is the identity. Based on this and the central limit theorem for an i.i.d standard Gaussian distribution, we can formulate the representational dynamics for each mask patch $y_j$ under the $\mathcal{L}$ as

$$
\begin{aligned}
\frac{d\hat{\mathbf{z}}_{y_j}}{dt} &= -\eta\hat{\Theta}_t(\boldsymbol{y}_j, \mathcal{Y})\nabla_{\hat{\mathcal{Z}}_y}\mathcal{L} \\
&= -\eta\hat{\Theta}_t(\boldsymbol{y}_j, \boldsymbol{y}_j)\nabla_{\hat{\mathbf{z}}_{y_j}}\mathcal{L} - \eta\sum_{q \neq j}\hat{\Theta}_t(\boldsymbol{y}_j, \boldsymbol{y}_q)\nabla_{\hat{\mathbf{z}}_{y_q}}\mathcal{L} \\
&= -\eta(\boldsymbol{y}_j^T\boldsymbol{y}_j)\nabla_{\hat{\mathbf{z}}_{y_j}}\mathcal{L} - \eta\sum_{q \neq j}(\boldsymbol{y}_j^T\boldsymbol{y}_q)\nabla_{\hat{\mathbf{z}}_{y_q}}\mathcal{L} \\
&= -\eta\nabla_{\hat{\mathbf{z}}_{y_j}}\mathcal{L} = -\eta\frac{\partial\mathcal{L}}{\partial\hat{\mathbf{z}}_{y_j}}(t) \\
&= -\eta\left(D_t\hat{\mathbf{z}}_{y_j} - \hat{\mathbf{z}}_{y_j}^a\right)D_t
\end{aligned}
\tag{13}
$$

where $D_t$ is a diagonal matrix composed of the eigenvalues $\lambda_k$. For $k$th entry of $\frac{d\hat{\mathbf{z}}_{y_j,k}}{dt}$, we can formulate the representational dynamics under the $\mathcal{L}$ as

$$
\begin{aligned}
\frac{d\hat{\mathbf{z}}_{y_j,k}}{dt} &= -\eta\nabla_{\hat{\mathbf{z}}_{y_j,k}}\mathcal{L} = -\eta\frac{\partial\mathcal{L}}{\partial\hat{\mathbf{z}}_{y_j,k}}(t) \\
&= -\eta\left(\lambda_k\hat{\mathbf{z}}_{y_j,k} - \hat{\mathbf{z}}_{y_j,k}\right)\lambda_k \\
&= \eta\lambda_k\left(\hat{\mathbf{z}}_{y_j,k}^a - \lambda_k\hat{\mathbf{z}}_{y_j,k}\right)
\end{aligned}
\tag{14}
$$

Since the expectation over the same masking blocks is $\hat{\mathbf{z}}_{y_j,k}$, we can have the dynamics for each mask patch $y_j$ as

$$
\frac{d\hat{\mathbf{z}}_{y_j,k}}{dt} = \eta\lambda_k(1 - \lambda_k)\hat{\mathbf{z}}_{y_j,k}
\tag{15}
$$

## B   Representation learning dynamics without the stop-grad and predictor

### B.1   Learning dynamics of the loss $\mathcal{L}$ without the stop-grad

Recall we consider linear predictor $W_P \in \mathbb{R}^{M \times M}$ in I-JPEA, the masking objective with the average $\ell_2$ distance between the predicted patch-level representations $\mathbf{z}_{y_j}$ and the target patch-level representations $\mathbf{z}_{y_j}^a$ as

$$
\mathcal{L} = \frac{1}{|M|}\sum_{i=1}^{M}\sum_{j \in \mathcal{B}_i}||W_P\mathbf{z}_{y_j} - \text{SG}(\mathbf{z}_{y_j}^a)||_2^2,
\tag{16}
$$

When removing the stop-grad operator in $\mathcal{L}$, we can have the masking objective as

$$
\begin{aligned}
\mathcal{L} &= \frac{1}{|M|}\sum_{i=1}^{M}\sum_{j \in \mathcal{B}_i}||W_P\mathbf{z}_{y_j} - \mathbf{z}_{y_j}^a||_2^2 \\
&= \frac{1}{|M|}\sum_{i=1}^{M}\sum_{j \in \mathcal{B}_i}\sum_{k}^{K}||\lambda_k\hat{\mathbf{z}}_{y_j,k} - (\hat{\mathbf{z}}_{y_j,k}^a)||_2^2
\end{aligned}
\tag{17}
$$

The representational dynamics for each mask patch $y_j$ under the $\mathcal{L}$ is formulated as

$$
\begin{aligned}
\frac{d\hat{\mathbf{z}}_{y_j,k}}{dt} &= -\eta \frac{\partial \mathcal{L}}{\partial \hat{\mathbf{z}}_{y_j,k}}(t) \\
&= -\eta \left( \frac{\partial \mathcal{L}}{\partial \hat{\mathbf{z}}_{y_j,k}}(t) + \frac{\partial \mathcal{L}}{\partial \hat{\mathbf{z}}_{y_j,k}^a}(t) \right) \\
&= -\eta \left( (\lambda_k \hat{\mathbf{z}}_{y_j,k} - \hat{\mathbf{z}}_{y_j,k}^a)\lambda_k + (\lambda_k \hat{\mathbf{z}}_{y_j,k} - \hat{\mathbf{z}}_{y_j,k}^a) \right) \\
&= -\eta(\lambda_k \hat{\mathbf{z}}_{y_j,k} - \hat{\mathbf{z}}_{y_j,k}^a)(\lambda_k - 1)
\end{aligned}
\tag{18}
$$

Since the expectation over the same masking blocks is $\hat{\mathbf{z}}_{y_j,k}$, we can have the dynamics for each mask patch $y_j$ as

$$
\frac{d\hat{\mathbf{z}}_{y_j,k}}{dt} = -\eta(1 - \lambda_k)^2 \hat{\mathbf{z}}_{y_j,k}
\tag{19}
$$

where $d\hat{\mathbf{z}}_{y_j,k}$ and $\hat{\mathbf{z}}_{y_j,k}$ always have opposite signs and the learned representations will become zero with exponentially decaying eigenvalues to collapse.

### B.2 Learning dynamics of the loss $\mathcal{L}$ without the predictor

Without the predictor $W_P$, we can have the masking objective as

$$
\begin{aligned}
\mathcal{L} &= \frac{1}{|M|} \sum_{i=1}^{M} \sum_{j \in \mathcal{B}_i} ||\mathbf{z}_{y_j} - \text{SG}(\mathbf{z}_{y_j}^a)||_2^2 \\
&= \frac{1}{|M|} \sum_{i=1}^{M} \sum_{j \in \mathcal{B}_i} \sum_{k}^{K} ||\hat{\mathbf{z}}_{y_j,k} - \text{SG}(\hat{\mathbf{z}}_{y_j,k}^a)||_2^2
\end{aligned}
\tag{20}
$$

In this case, we have $\lambda_k = 1$ for all independent eigenvalues in the eigenbasis. Based on the representation dynamics for each mask patch $y_j$ as $\dfrac{d\hat{\mathbf{z}}_{y_j,k}}{dt} = \eta\lambda_k(1 - \lambda_k)\hat{\mathbf{z}}_{y_j,k}$, we will have $\dfrac{d\hat{\mathbf{z}}_{y_j,k}}{dt} = 0$, and thus the representations will not be updated in the end.

## C  Algorithm

In this section, we provide detailed algorithmic and implementation aspects of the Contrastive-JEPA (C-JEPA) framework, elaborating on the loss functions, pseudo-code, and experimental setup used throughout our studies. These details aim to enable reproducibility and deeper understanding of the modifications introduced in the C-JEPA compared to its predecessors.

### C.1  Loss

The loss function for C-JEPA is a combination of the original JEPA loss and the VICReg-inspired regularization terms, structured as follows:

$$
\mathcal{L} = \mathcal{L}_{JEPA} + \beta_{vicreg} * \mathcal{L}_{VICReg}
\tag{21}
$$

where $\beta vicreg = 0.001$ provides a scaling factor to balance the original JEPA loss with the new regularization terms introduced from VICReg. The VICReg loss component is defined as:

$$
\mathcal{L}_{VICReg} = \beta_{sim} * \mathcal{L}_{sim} + \beta_{std} * \mathcal{L}_{std} + \beta_{cov} * \mathcal{L}_{cov}
\tag{22}
$$

with $\beta_{sim} = 25$, $\beta_{std} = 25$, and $\beta_{cov} = 1$, which scale the similarity, standard deviation, and covariance regularization terms respectively, integral to maintaining the diversity and stability of the learned representations.

## C.2 Pseudo Code

The following pseudo-code outlines the batch-wise application of VICReg within the C-JEPA framework, providing clarity on the computational steps involved during training:

```
# Pseudo-code for batch-wise VICReg application in C-JEPA
for each batch in dataset:
    # Forward pass through the context and target encoders
    context_embeddings, target_embeddings = encode(batch)

    # Apply VICReg terms
    sim_loss = compute_similarity_loss(context_embeddings, target_embeddings)
    std_loss = compute_standard_deviation_loss(context_embeddings)
    cov_loss = compute_covariance_loss(context_embeddings)

    # Combine losses
    vicreg_loss = beta_sim * sim_loss + beta_std * std_loss + beta_cov * cov_loss
    total_loss = jepa_loss(batch) + beta_vicreg * vicreg_loss

    # Backward pass and update weights
    optimizer.zero_grad()
    total_loss.backward()
    optimizer.step()
```

Algorithm 1 outlines the procedure for applying VICReg in a batch-wise manner within the C-JEPA framework, focusing on multiple views in the context of contrastive learning.

Table 5: **Pretraining setting for ViT-T**. All models trained for 100 and 600 epochs.

| config | value |
|---|---|
| optimizer | AdamW [48] |
| epochs | 100 or 600 |
| learning rate | $1e^{-3}$ |
| weight decay | $(0.04, 0.4)$ |
| batch size | 2048 |
| learning rate schedule | cosine decay [49] |
| warmup epochs | 15 |
| encoder arch. | ViT-T |
| predicted targets | 4 |
| predictor depth | 6 |
| predictor attention heads | 12 |
| predictor embedding dim. | 192 |

Table 6: **Pretraining setting for ViT-S**. All models trained for 100 epochs.

| config | value |
|---|---|
| optimizer | AdamW [48] |
| epochs | 100 |
| learning rate | $8e^{-4}$ |
| weight decay | $(0.04, 0.4)$ |
| batch size | 2048 |
| learning rate schedule | cosine decay [49] |
| warmup epochs | 15 |
| encoder arch. | ViT-S |
| predicted targets | 4 |
| predictor depth | 6 |
| predictor attention heads | 12 |
| predictor embedding dim. | 384 |

**Algorithm 1** VICReg loss for our framework

---

**Require:** $zc$: Context embeddings tensor of size [batch_size, num_block, num_mask_patch, dimension]

**Ensure:** $loss$: Computed loss for the batch

1: **function** VICREGLOSS($zc$, args)
2:     $zc \leftarrow$ MEAN($zc$, dim $= 2$)                          ▷ Average over the mask patches
3:     $zc \leftarrow$ PROJECTOR($zc$)                     ▷ Apply projection to the embeddings
4:     **if** 'vicreg' in args **then**
5:         $criterion \leftarrow$ vicreg_criterion
6:     **else**
7:         $criterion \leftarrow$ CosineSimilarity(dim=1)
8:     **end if**
9:     $loss \leftarrow 0$
10:    $num\_masks \leftarrow$ SHAPE($zc$)[1]
11:    **for** $i \leftarrow 0$ **to** $num\_masks - 1$ **do**
12:        **for** $j \leftarrow 0$ **to** $num\_masks - 1$ **do**
13:            **if** $i \neq j$ **then**
14:                $zc_i \leftarrow zc[:, i, :]$
15:                $zc_j \leftarrow zc[:, j, :]$
16:            **if** 'vicreg' in args **then**
17:                $zc\_gather \leftarrow$ FULLGATHERLAYER($zc$)
18:                $zc\_gather_i \leftarrow zc\_gather[:, i, :]$
19:                $zc\_gather_j \leftarrow zc\_gather[:, j, :]$
20:                $loss \leftarrow loss +$ CRITERION($zc_i, zc_j, zc\_gather_i, zc\_gather_j$)
21:            **else**
22:                $loss \leftarrow loss -$ CRITERION($zc_i, zc_j$)
23:            **end if**
24:            **end if**
25:        **end for**
26:    **end for**
27:    $loss \leftarrow loss / (num\_masks \times (num\_masks - 1))$
28:    **return** $loss$
29: **end function**

---

Table 7: **Pretraining setting for ViT-B**. All models trained for 600 epochs.

| config | value |
|---|---|
| optimizer | AdamW [48] |
| epochs | 600 |
| learning rate | $8e^{-4}$ |
| weight decay | $(0.04, 0.4)$ |
| batch size | 2048 |
| learning rate schedule | cosine decay [49] |
| warmup epochs | 15 |
| encoder arch. | ViT-B |
| predicted targets | 4 |
| predictor depth | 6 |
| predictor attention heads | 12 |
| predictor embedding dim. | 384 |

Table 8: **Pretraining setting for ViT-L**. Trained for 600 epochs.

| config | value |
|---|---|
| optimizer | AdamW [48] |
| epochs | 600 |
| learning rate | $8e^{-4}$ |
| weight decay | $(0.04, 0.4)$ |
| batch size | 2048 |
| learning rate schedule | cosine decay [49] |
| warmup epochs | 15 |
| encoder arch. | ViT-L |
| predicted targets | 4 |
| predictor depth | 12 |
| predictor attention heads | 16 |
| predictor embedding dim. | 384 |

## C.3 Implementation Details

For a comprehensive implementation, we detail the specific settings used during our experiments:

- **Image Preprocessing:** All input images were resized to a resolution of $224 \times 224$ pixels.

- **Patch Size:** Following prior work, a patch size of $16 \times 16$ pixels was used.

Table 9: **ImageNet-1K image classification.** We perform a linear evaluation on pre-trained ViT-T/16 and ViT-S/16 models for image classification on ImageNet-1K benchmark. We report the Top-1 accuracies using knn, linprob and fine-tune settings to evaluate the quality of pre-trained representations. The best results are indicated in **bold**.

| Method | Backbone | Epochs | knn | linprob | fine-tune |
|---|---|---|---|---|---|
| I-JEPA [2] | ViT-T/16 | 100 | 15.16 | 22.13 | 64.46 |
| | ViT-S/16 | 100 | 24.36 | 29.61 | 76.87 |
| C-JEPA (ours) | ViT-T/16 | 100 | **19.18** | **23.31** | **66.77** |
| | ViT-S/16 | 100 | **26.85** | **32.35** | **78.17** |

Table 10: **COCO object detection and instance segmentation.** We fine-tuned pre-trained ViT-T/16 and ViT-S/16 models to perform COCO object detection and instance segmentation. The $AP^{box}$ and $AP^{mask}$ metrics denote the results of COCO detection, and COCO segmentation, respectively. The best results are indicated in **bold**.

| Method | Backbone | Pretrain Epoch | $AP^{bb}$ | $AP^{bb}_{50}$ | $AP^{bb}_{75}$ | $AP^{mk}$ | $AP^{mk}_{50}$ | $AP^{mk}_{75}$ |
|---|---|---|---|---|---|---|---|---|
| I-JEPA [2] | ViT-T/16 | 100 | 30.5 | 52.6 | 32.8 | 29.1 | 49.5 | 30.6 |
| | ViT-S/16 | 100 | 36.9 | 58.5 | 39.2 | 35.2 | 56.3 | 37.3 |
| C-JEPA (ours) | ViT-T/16 | 100 | **32.8** | **54.5** | **34.6** | **30.9** | **51.4** | **32.2** |
| | ViT-S/16 | 100 | **38.5** | **59.7** | **40.3** | **37.1** | **57.5** | **39.2** |

- **Vision Transformer Configurations:** We utilized tiny, small, base, and large models of the ViT architecture, with specific settings for each model scale provided in Tables 5, 6, 7, and 8.
- **Optimizer:** The AdamW optimizer was used for training, with specific learning rate schedules and weight decay settings described.
- **Batch Size and Training Schedule:** A default batch size of 2048 was used, with a learning rate and momentum adjustment schedule to optimize convergence.

To validate the efficacy of C-JEPA, we employed several benchmark datasets and tasks:

- **Image Classification on ImageNet-1K [36]**
- **Object Detection and Instance Segmentation on MS-COCO [37]**
- **Semantic Segmentation on ADE20K [38, 39]**
- **Video Object Segmentation on DAVIS-2017 [50]**
- **Low-Level Tasks on Clevr Dataset [45]**

For each task, we adopted established methodologies and frameworks, such as Mask R-CNN [42] for object detection and Semantic FPN and UPerNet [44] for semantic segmentation, ensuring that our findings were comparable with state-of-the-art results.

## D    More Experiments

In this part, we provide additional experimental analyses to validate the effectiveness of the proposed C-JEPA framework further. These experiments encompass various downstream tasks, ablation studies, and the impact of different components and coefficients in the model, specifically focusing on the Variance, Covariance, and Invariance terms from VICReg across different configurations of Vision Transformers.

**Downstream Results on ViT-T and ViT-S.** We evaluated the performance of pre-trained ViT-T/16 and ViT-S/16 models across a series of downstream tasks to assess the quality of the representations learned by the C-JEPA framework. For image classification, we performed linear evaluations using knn, linear probing (linprob), and fine-tuning methods: The results of top-1 accuracies, presented in Table 9, indicate the robustness and quality of the features extracted by our pre-trained models

Table 11: **ADE20K semantic segmentation.** We fine-tuned pre-trained ViT-T/16 and ViT-S/16 models to perform ADE20K semantic segmentation. The mIoU, aAcc, and mAcc metrics denote the results of ADE20K segmentation. The best results are indicated in **bold**.

| Method | Backbone | Pretrain Epoch | Semantic FPN | | | UPerNet | | |
|---|---|---|---|---|---|---|---|---|
| | | | mIoU | aAcc | mAcc | mIoU | aAcc | mAcc |
| I-JEPA [2] | ViT-T/16 | 100 | 21.56 | 72.28 | 30.26 | 33.15 | 75.36 | 43.05 |
| | ViT-S/16 | 100 | 27.17 | 73.64 | 37.11 | 36.98 | 78.30 | 46.66 |
| C-JEPA (ours) | ViT-T/16 | 100 | **24.40** | **73.56** | **32.88** | **34.44** | **76.96** | **44.56** |
| | ViT-S/16 | 100 | **31.36** | **76.27** | **41.34** | **38.68** | **79.07** | **48.86** |

Table 12: **Ablation studies on component analysis for faster convergence.** We perform ablation studies on Variance/Covariance and Invariance modules in VICReg using ViT-T/16 model. The best results are indicated in **bold**.

| I-JEPA | Variance/Covariance | Invariance | Backbone | Pretrain Epoch | knn | linprob | fine-tune |
|---|---|---|---|---|---|---|---|
| ✓ | ✗ | ✗ | ViT-T/16 | 100 | 15.16 | 22.15 | 64.46 |
| ✓(mean) | ✓(collapse) | ✗ | ViT-T/16 | 100 | 18.75 | 22.92 | 66.15 |
| ✓ | ✓ | ✓ | ViT-T/16 | 100 | **19.18** | **23.39** | **66.77** |
| ✓(EMA for collapse) | ✗ | ✓(mean) | ViT-T/16 | 100 | 18.57 | 22.57 | 65.92 |

under different evaluation settings. For object detection and instance segmentation, we fine-tuned the pre-trained models to perform tasks on the COCO dataset. The average precision for bounding box detection (AP$^{\texttt{box}}$) and mask segmentation (AP$^{\texttt{mask}}$) are reported in Table 10, demonstrating the applicability of C-JEPA pre-trained models in complex vision tasks. The models were also fine-tuned for semantic segmentation tasks, and metrics such as mean Intersection over Union (mIoU), average class accuracy (aAcc), and mean accuracy (mAcc) are detailed in Table 11, highlighting the effectiveness of our approach in handling pixel-level classification tasks.

**Variance/Covariance and Invariance in VICReg.** To dissect the contributions of individual VICReg components within the C-JEPA, we conducted focused ablation studies using the ViT-T/16 model. For faster convergence, initial ablations in Table 12 analyze the impact of the Variance/Covariance and Invariance modules on speeding up the model training. Further studies on better convergence, shown in Table 13, investigate how these modules contribute to achieving better training dynamics and model performance.

**VICReg and Invariance Coefficients.** For VICReg coefficients $\beta_{vicreg}$, we varied the coefficients to understand their influence on model training and final performance, with results compiled in Table 14. The impact of varying the invariance coefficient $\beta_{sim}$ is explored to optimize the balance between similarity and variance, with findings in Table 15.

These experiments underline the nuanced contributions of the VICReg components to the overall efficacy of the C-JEPA framework. By systematically varying key parameters and assessing their impacts across a broad spectrum of tasks, we establish a comprehensive understanding of the model's behavior and performance characteristics. This rigorous empirical foundation supports the theoretical enhancements proposed in C-JEPA, affirming its potential to advance the state-of-the-art in unsupervised visual representation learning.

# E  More Visualizations

In order to provide a comprehensive understanding of the advancements offered by the C-JEPA framework, we include a detailed visual analysis alongside our quantitative evaluations. This section of the appendix is dedicated to showcasing attention visualizations that illustrate the practical impacts of the theoretical improvements achieved through the integration of VICReg components within the C-JEPA framework.

To better understand how the C-JEPA model processes visual information, we utilize attention visualization techniques. These techniques highlight the model's ability to focus on relevant features within an image, which is crucial for tasks involving fine-grained recognition and localization. The following figures 4, 5, 6, 7, and 8 provide a visual comparison between the base I-JEPA and the

Table 13: **Ablation studies on component analysis for better convergence.** We perform ablation studies on Variance/Covariance and Invariance modules in VICReg using ViT-T/16 model. The best results are indicated in **bold**.

| I-JEPA | Variance/Covariance | Invariance | Backbone | Pretrain Epoch | knn | linprob | fine-tune |
|---|---|---|---|---|---|---|---|
| ✓ | ✗ | ✗ | ViT-T/16 | 600 | 20.53 | 26.83 | 68.68 |
| ✓(mean) | ✓(collapse) | ✗ | ViT-T/16 | 600 | 25.87 | 29.65 | 71.06 |
| ✓ | ✓ | ✓ | ViT-T/16 | 600 | **27.61** | **31.86** | **71.52** |
| ✓(EMA for collapse) | ✗ | ✓(mean) | ViT-T/16 | 600 | 24.95 | 27.92 | 70.58 |

Table 14: **Ablation studies on component analysis for VICReg coefficients.** We perform ablation studies on VICReg coefficients $(\cdot)$ using ViT-T/16 model. The best results are indicated in **bold**.

| VICReg $\beta_{vicreg}$ | Backbone | Pretrain Epoch | $(\mathcal{J}\&\mathcal{F})_m$ | $\mathcal{J}_m$ | $\mathcal{F}_m$ |
|---|---|---|---|---|---|
| 0.0001 | ViT-T/16 | 100 | 43.6 | 43.0 | 44.3 |
| 0.0002 | ViT-T/16 | 100 | 45.7 | 45.3 | 46.1 |
| 0.0005 | ViT-T/16 | 100 | 47.3 | 46.8 | 47.8 |
| 0.001 | ViT-T/16 | 100 | **50.2** | **49.8** | **50.6** |
| 0.005 | ViT-T/16 | 100 | 46.0 | 45.4 | 46.6 |
| 0.01 | ViT-T/16 | 100 | 31.9 | 33.1 | 30.7 |
| 0.02 | ViT-T/16 | 100 | 29.5 | 30.6 | 28.4 |
| 0.05 | ViT-T/16 | 100 | 24.4 | 25.8 | 23.1 |
| 0.1 | ViT-T/16 | 100 | Collapsing | | |

enhanced C-JEPA models. For each sample, columns denote the original image, attention maps from the target encoder in I-JEPA, attention maps from the target encoder in our C-JEPA, and attention maps from the context encoder in our C-JEPA.

These attention visualizations demonstrate the qualitative improvements in the C-JEPA's ability to focus and contextualize features compared to the base I-JEPA model. The C-JEPA model produces more focused and contextually relevant attention maps, which is a direct result of the enhanced model architecture that incorporates VICReg's variance, covariance, and invariance regularization strategies. The enhanced attention capabilities of C-JEPA facilitate better feature extraction from complex visual inputs, which is critical for improving performance on downstream tasks. The more detailed and precise attention maps correlate strongly with the theoretical enhancements integrated into the C-JEPA framework, such as improved handling of variance and invariance across different visual domains.

The attention visualizations provided in this section validate the theoretical improvements posited in the earlier sections of this paper. They offer qualitative evidence that supports the quantitative results, underscoring the benefits of the C-JEPA model in capturing more nuanced and robust representations. Overall, these visualizations highlight the C-JEPA's ability to maintain stable and meaningful patch-level representations across a variety of image contexts, marking a significant advancement over its predecessor.

Table 15: **Ablation studies on component analysis for invariance coefficients.** We perform ablation studies on invariance coefficients (·) using ViT-T/16 model. The best results are indicated in **bold**.

| I-JEPA | Variance/Covariance | Invariance | Backbone | Pretrain Epoch | knn | linprob | fine-tune |
|:---:|:---:|:---:|:---:|:---:|:---:|:---:|:---:|
| ✓ | ✗ | ✓(25) | ViT-T/16 | 100 | 18.03 | 21.72 | 65.68 |
| ✓ | ✗ | ✓(20) | ViT-T/16 | 100 | 18.35 | 22.25 | 65.73 |
| ✓ | ✗ | ✓(15) | ViT-T/16 | 100 | **18.57** | **22.53** | **65.92** |
| ✓ | ✗ | ✓(10) | ViT-T/16 | 100 | 18.45 | 22.36 | 65.81 |

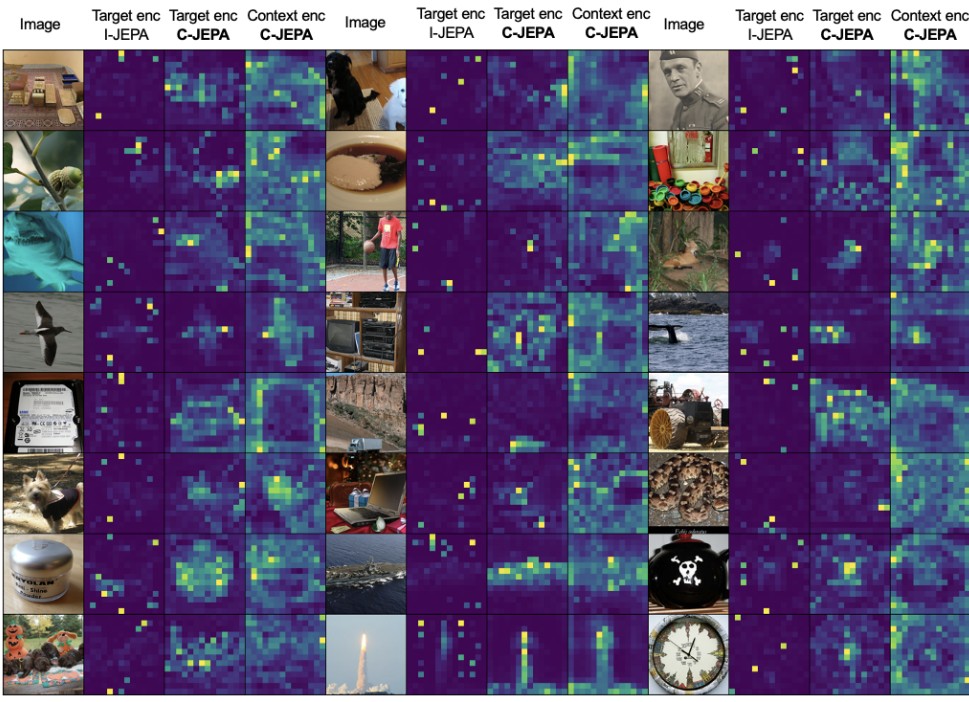

Figure 4: **Qualitative visualization of learned attention maps using ViT-B/16 model.** Columns for each sample denote the original image, attention maps from the target encoder in I-JEPA [2], attention maps from the target encoder in our C-JEPA, and attention maps from the context encoder in our C-JEPA. Our C-JEPA achieves much better attention maps.

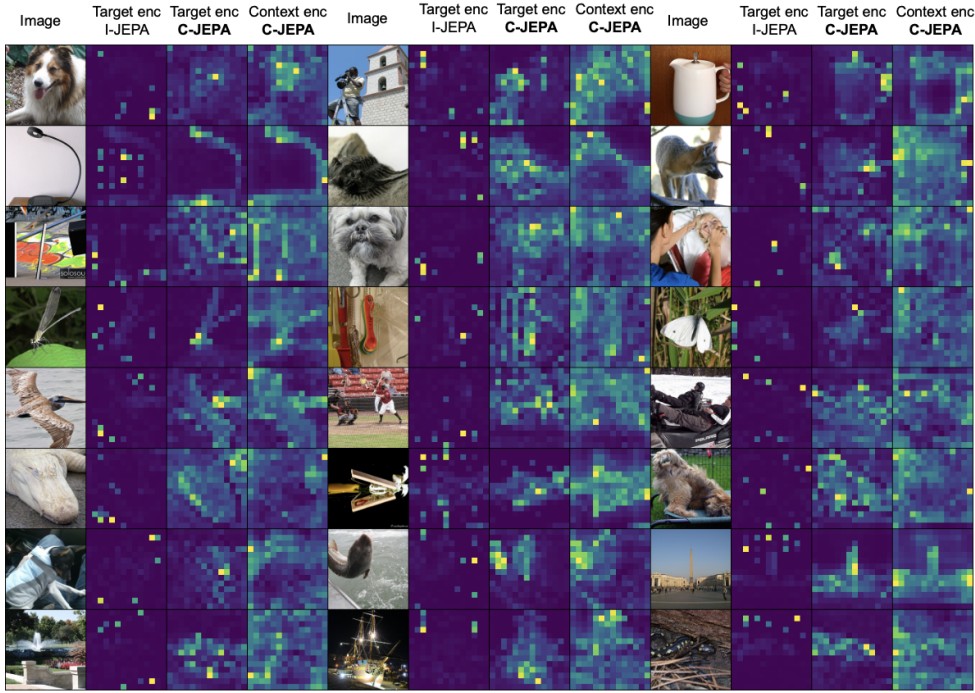

Figure 5: **Qualitative visualization of learned attention maps using ViT-B/16 model.** Columns for each sample denote the original image, attention maps from the target encoder in I-JEPA [2], attention maps from the target encoder in our C-JEPA, and attention maps from the context encoder in our C-JEPA. Our C-JEPA achieves much better attention maps.

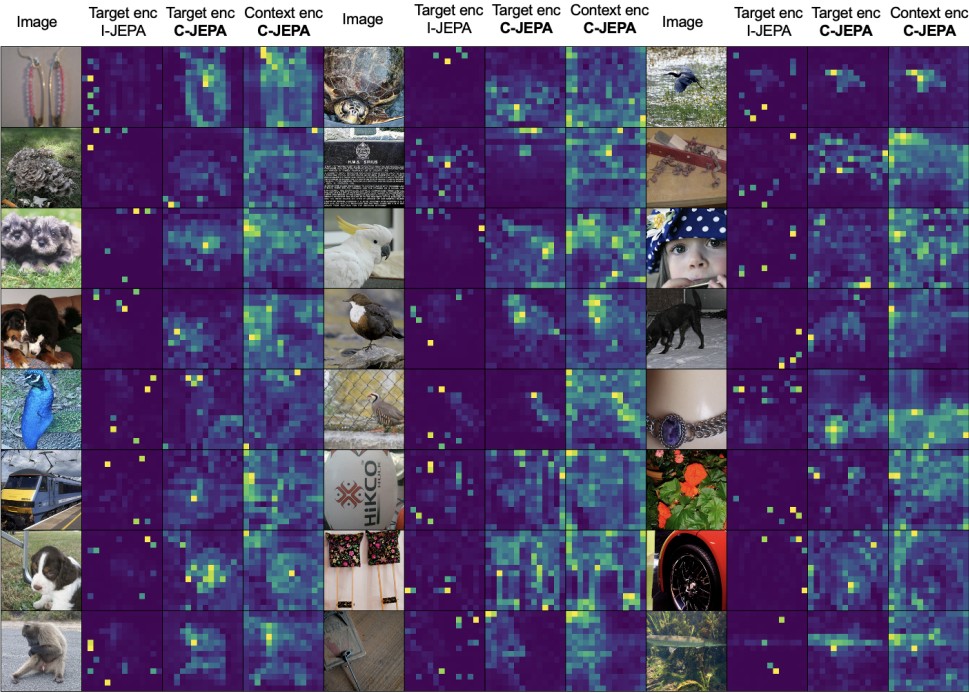

Figure 6: **Qualitative visualization of learned attention maps using ViT-B/16 model.** Columns for each sample denote the original image, attention maps from the target encoder in I-JEPA [2], attention maps from the target encoder in our C-JEPA, and attention maps from the context encoder in our C-JEPA. Our C-JEPA achieves much better attention maps.

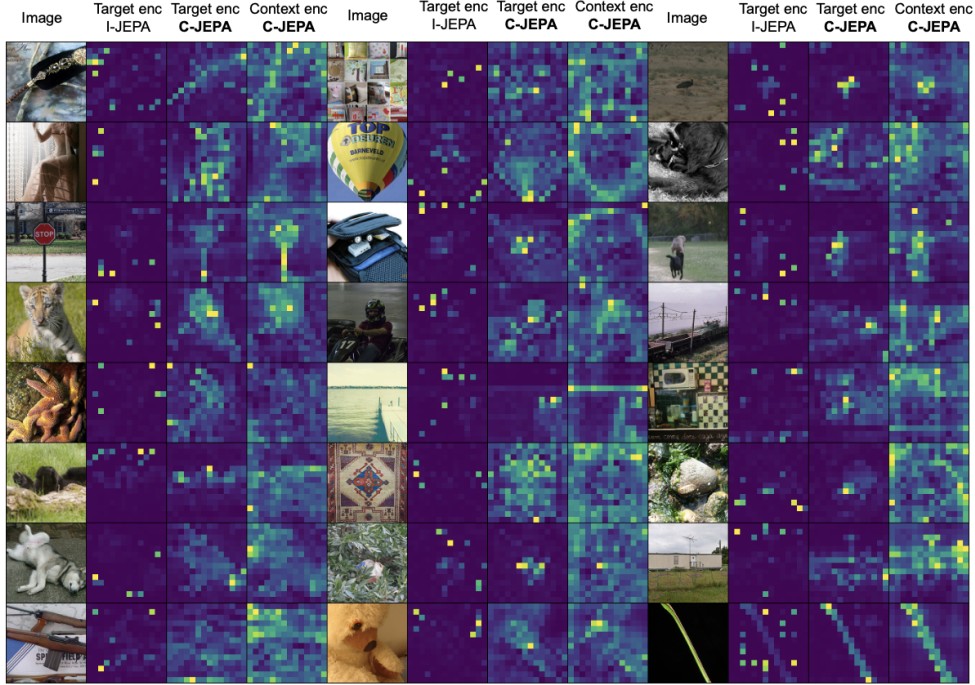

Figure 7: **Qualitative visualization of learned attention maps using ViT-B/16 model.** Columns for each sample denote the original image, attention maps from the target encoder in I-JEPA [2], attention maps from the target encoder in our C-JEPA, and attention maps from the context encoder in our C-JEPA. Our C-JEPA achieves much better attention maps.

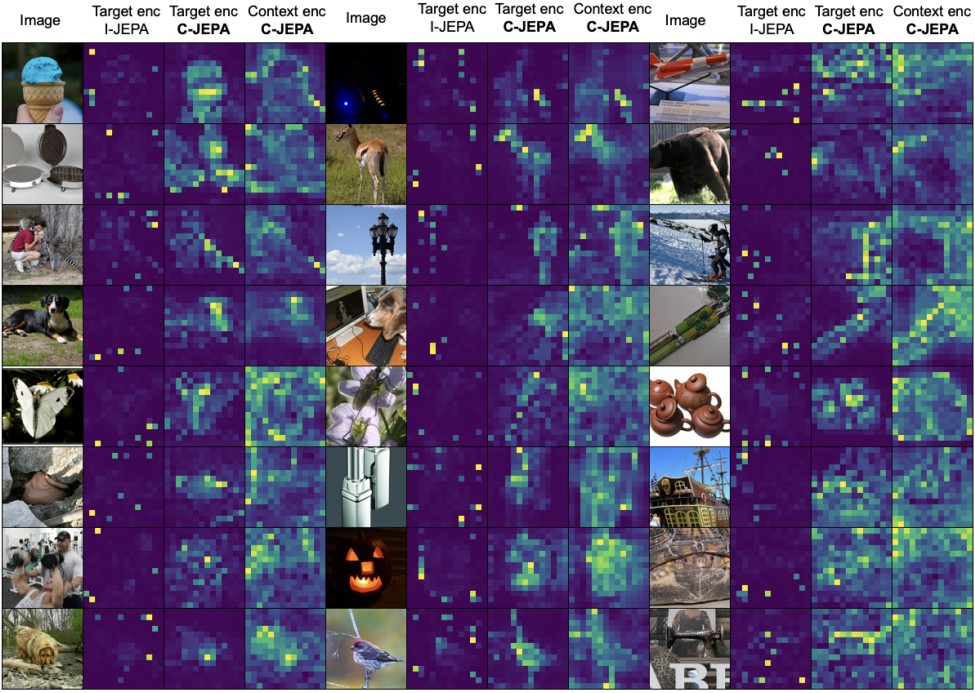

Figure 8: **Qualitative visualization of learned attention maps using ViT-B/16 model.** Columns for each sample denote the original image, attention maps from the target encoder in I-JEPA [2], attention maps from the target encoder in our C-JEPA, and attention maps from the context encoder in our C-JEPA. Our C-JEPA achieves much better attention maps.

