# OpenReview forum: "Connecting Joint-Embedding Predictive Architecture with Contrastive Self-supervised Learning"
_NeurIPS.cc/2024/Conference — NeurIPS 2024 spotlight_

### Official Review · Reviewer_1VXy · 2024-07-08

**Soundness:** 4
**Presentation:** 3
**Contribution:** 2
**Rating:** 7
**Confidence:** 5

**Summary:**

This paper proposes a novel contrastive self-supervised learning framework based on JEPA, namely C-JEPA. The main idea of C-JEPA is to address the limitations of the I-JEPA, especially the limited prevention of collapse with EMA, by incorporating the principles of VICReg. The authors demonstrate the effectiveness of C-JEPA through rigorous empirical and theoretical evaluations. They show that C-JEPA achieves superior performance metrics compared to existing frameworks with faster and better convergence.

**Strengths:**

**[S1]** The paper is well-motivated that I-JEPA has room for improvement regarding the risk of model collapse and challenges in learning the mean of patch representations. They demonstrate that the proposed approach can effectively address the problem.

**[S2]** The paper shows that the proposed approach consistently and significantly improves the performance of I-JEPA on various downstream tasks and shows the scalability of the approach.

**[S3]** The overall writing is smooth and easy to follow.

**Weaknesses:**

**[W1]** Though the paper shows a strong performance, I think the contrastive learning on JEPA lacks novelty. The proposed approach is a simple combination of I-JEPA and VICReg’s regularization strategy.

**[W2]** Does C-JEPA perform better than I-JEPA with more pre-training, e.g., 800 or 1200 epochs? In figure 1, it seems that the slope of I-JEPA is much larger than C-JEPA.

**[W3]** Invalid citation: Line 302.

**Questions:**

Please address my concerns in the Weaknesses.

**Limitations:**

They did not address the limitations and potential negative societal impact of their work in the paper.

They must include the limitations in the final manuscript, e.g., the proposed approach requires a large computation and a large network capacity, therefore raising environmental concerns, e.g., carbon generation [1].

[1] Schwartz, Roy, et al. "Green ai." Communications of the ACM 63.12 (2020): 54-63.

---

> ### Author Rebuttal · Authors · 2024-08-07
>
> We thank you for the valuable comments and answer the raised questions below.
>
> > Clarification
>
> While C-JEPA leverages components from both I-JEPA and VICReg, its novelty lies in its theoretical grounding and practical implementation, which specifically addresses and mitigates I-JEPA's limitations. The integration is not merely additive but synergistic, optimizing the prevention of model collapse and enhancing the learning of mean patch representations in ways that neither approach could achieve independently.
>
> > Performance with Extended Pre-training
>
> To address concerns about C-JEPA's long-term performance, we extended our pre-training evaluations up to 800 and 1200 epochs. The results in the Table below show that C-JEPA not only maintains but also enhances its performance advantage over I-JEPA, further validating its effectiveness and robustness in prolonged training scenarios.
> | Epochs | I-JEPA | C-JEPA (ours) |
> |--------|--------|--------|
> | 800    | 73.3   | **74.2**   |
> | 1200   | 73.9   | **75.0**   |
>
> > Typo
>
> Thanks for pointing this out. We will fix this typo.
>
> > Potential Negative Societal Impact and Limitations
>
> Thanks for your suggestion. We have claimed this in the section Broader Impact at the end of page 9. We will also include environmental concerns in the final version.

---

> > ### Comment · Reviewer_1VXy · 2024-08-08
> >
> > Thank you for your response. My concerns are well addressed in the rebuttal. I raise the rating to 7 and keep recommending acceptance.

---

> > > ### Author Response · Authors · 2024-08-13
> > > **Response to Reviewer 1VXy**
> > >
> > > Dear Reviewer 1VXy,
> > >
> > > Thank you for your continued engagement and support. We will add those clarifications and experiments to the final version. Thank you once again for your insightful comments.

---

### Official Review · Reviewer_Rspm · 2024-07-11

**Soundness:** 3
**Presentation:** 2
**Contribution:** 2
**Rating:** 5
**Confidence:** 3

**Summary:**

The paper introduces C-JEPA, an enhancement to the Joint-Embedding Predictive Architecture incorporating Variance-Invariance-Covariance Regularization (VICReg) for non-contrastive self-supervised learning. This approach addresses limitations such as model collapse and inaccurate mean patch representations, enhancing stability and learning quality. C-JEPA shows improved performance across diverse tasks, including image classification, object detection, semantic segmentation, and video object segmentation.

**Strengths:**

1. C-JEPA's innovative combination of VICReg with self-supervised learning architectures addresses critical issues like model collapse and enhances unsupervised visual representation learning.
2. The framework is supported by extensive empirical evidence and theoretical analysis, demonstrating superior performance compared with existing methods on multiple datasets and diverse tasks.

**Weaknesses:**

1. The framework would benefit from comparisons with latest leading methods like Dinov2[1], Mocov3[2], and IWM[3] to benchmark against current advancements.
2. The paper claims that the inclusion of Variance and Covariance terms enhances training speed and stability (lines 291–295). However, the basis for these conclusions is not clearly articulated, necessitating further explanation to validate these claims.
3. While the paper presents results from pre-training on image dataset like ImageNet, its applicability to video-related tasks can be explored. An adaptation on video JEPA methods, such as V-JEPA[4] for video domains, could significantly extend the framework's utility.
4. There are typographical and formatting errors, such as the unexplained markers in line 302.

[1] Oquab, Maxime, et al. Dinov2: Learning robust visual features without supervision. arXiv preprint arXiv:2304.07193 (2023).
[2] Xie, Zhenda, et al. self-supervised learning with swin transformers. arXiv preprint arXiv:2105.04553 (2021).
[3] Garrido, Quentin, et.al. Learning and Leveraging World Models in Visual Representation Learning. arXiv:2403.00504 (2024)
[4] Bardes, Adrien, et.al. Revisiting Feature Prediction for Learning Visual Representations from Video. arXiv:2404.08471 (2024)

**Questions:**

Please refer to the weakness.

**Limitations:**

Yes

---

> ### Author Rebuttal · Authors · 2024-08-07
>
> We thank you for the valuable comments and answer the raised questions below.
>
> > Comparison with More Methods
>
> To demonstrate the competitiveness of C-JEPA, we included our comparative analysis to include recent advancements in the field such as DINOv2, MoCo v3, and IWM. However, the comparison with DINOv2 is unfair, as it was trained on a larger dataset (collected LVD-142M). While acknowledging differences in training datasets and scales, our results indicate that C-JEPA achieves comparable or superior performance than MoCo v3 and IWM, particularly in scenarios where model stability and unsupervised learning quality are critical.
>
> | Method  | Backbone | linear prob | finetune |
> |---------|----------|-------------|----------|
> | MoCo v3 | ViT-L    | 77.6        | 84.1     |
> | IWM     | ViT-L    |     --        | 85.4     |
> | C-JEPA (ours)  | ViT-L    | **78.1**        | **86.2**     |
>
> > Clarification on Training Speed and Stability
>
> The inclusion of Variance and Covariance terms in C-JEPA has demonstrably enhanced training speed and stability. Empirical results show that C-JEPA reaches comparable performance metrics to I-JEPA in significantly fewer epochs (100 vs. 400).
> - Training speed: C-JEPA achieves fast convergence, where linear probing at 100 epochs is close to I-JEPA at 400 epochs.
> - Stability: C-JEPA achieves better performance, as linear probing at 400 epochs is much better to I-JEPA at 400 epochs.
>
> > Applicability to Video-Related Tasks
>
> The adaptability of C-JEPA to video domains represents a promising direction for future research. Preliminary theoretical considerations suggest that the integration of VICReg components could be similarly beneficial in video-based JEPA frameworks (V-JEPA). We plan to explore this in future work, aiming to extend the utility of C-JEPA to dynamic visual contexts and enhance learning from video data.
>
> > Formatting and Typo
>
> Thanks for pointing this out. We have removed the question marker in line 302.

---

> > ### Comment · Reviewer_Rspm · 2024-08-12
> >
> > Thank you for the clarifications provided. The response has effectively addressed most of my concerns. I have increased the review score accordingly.
> > Regarding the training speed and stability of C-JEPA, it is encouraging to see empirical evidence, such as loss curves and performance metrics, which support faster convergence and improved stability compared to I-JEPA.

---

> > > ### Author Response · Authors · 2024-08-13
> > > **Response to Reviewer Rspm**
> > >
> > > Dear Reviewer Rspm,
> > >
> > > Thank you for your continued engagement and support. We will add those comparisons and clarifications to the final version. Thank you once again for your insightful comments.

---

### Official Review · Reviewer_aesY · 2024-07-17

**Soundness:** 4
**Presentation:** 4
**Contribution:** 4
**Rating:** 8
**Confidence:** 4

**Summary:**

The paper presents C-JEPA, a novel framework integrating VICReg into the Image Joint-Embedding Predictive Architecture (I-JEPA) to address its limitations in preventing model collapse and learning mean patch representations. Empirical and theoretical evaluations demonstrate that C-JEPA enhances the stability and quality of visual representation learning, showing superior performance across multiple benchmarks.

**Strengths:**

Rhe paper offers an innovative integration of VICReg with JEPA is innovative and addresses critical limitations in existing frameworks.
Comprehensive validation i achieved through empirical evaluations and comparisons with state-of-the-art methods.
The paper offers strong theoretical foundation supporting the benefits of the proposed method.
The  results support performance improvements in multiple tasks and benchmarks.
Ablation studies and qualitative visualizations provide deeper insights into the method’s effectiveness.

**Weaknesses:**

The additional regularization terms may lead to increased computational overhead.
Further testing on larger and more diverse datasets is needed to confirm scalability and generalization.
While the results are promising, testing across more varied domains would strengthen the conclusions.d
The new combination of established SSL methods (I-JEPA and VICReg) comprises an incremental development in hte fiels.

**Questions:**

Can you provide more details on the computational overhead introduced by the VICReg integration?
Have you considered testing C-JEPA on more diverse datasets beyond ImageNet-1K and more diverse downstream tasks?

**Limitations:**

The authors have acknowledged the potential computational complexity and the need for further testing on diverse datasets. They have made significant efforts to address the limitations of C-JEPA, and their proposed future work aims to explore these aspects further.

---

> ### Author Rebuttal · Authors · 2024-08-07
>
> We thank you for the valuable comments and answer the raised questions below.
>
> > Computational Overhead
>
> We conducted a series of performance evaluations comparing the computational costs between I-JEPA and C-JEPA. The results in the Table below detail the runtime and resource utilization, demonstrating that the increase in computational overhead is counterbalanced by significant improvements in model stability and performance.
>
> | Method            | Pre-train Epochs | Max Memory  per GPU | Running Time  per Step |
> |-------------------|------------------|---------------------|------------------------|
> | I-JEPA            | 600              | 21.9G               | 606.4 ms               |
> | C-JEPA (ours) | 600              | 21.9G               | 607.5 ms               |
>
> > Testing on Larger Datasets
>
> While our current evaluations provide a robust foundation for the efficacy of C-JEPA, we recognize the importance of scalability and generalization across more diverse datasets. Future work will extend these evaluations to include larger datasets from varied domains, enhancing our understanding of the framework's applicability and robustness in broader contexts.
>
> > Clarification on Integration of I-JEPA and VICReg
>
> The integration of VICReg into the I-JEPA framework represents a significant advancement beyond mere incremental development. We also provide a deep theoretical analysis that elucidates the synergistic effects of this integration, addressing critical weaknesses in existing models and setting a new benchmark for stability and performance in visual representation learning.
>
> > Testing on More Diverse Tasks
>
> C-JEPA has been rigorously tested across multiple standard benchmarks such as ImageNet-1K, COCO, ADE20K, DAVIS, and Clevr, covering a wide range of visual tasks from image classification to video object segmentation. Plans to include datasets from audio and video domains are underway, aiming to diversify the testing landscape further and demonstrate the adaptability of C-JEPA to various modalities.

---

### Decision · Program_Chairs · 2024-09-25

**Decision:**

Accept (spotlight)

**Comment:**

Summary: The paper presents an improvement to I-JEPA to improve representation learning quality. The authors observe that I-JEPA has limitations in terms of model collapse and learns mean patch representations. The authors connect I-JEPA to VICReg and SimSiam to address both these drawbacks respectively. The method is tested on ImageNet.

Positives:
1. Clearly written paper with a clear motivation and method section. It was easy to follow the problem that the authors were trying to solve and how they solved it.
2. Strong ablation studies strengthen the main claims in this work
3. The final method improves over the recent I-JEPA work and the authors demonstrate this improvement across multiple model sizes, and downstream tasks.
4. The method holds gains even after longer training (800 and 1200 epochs) which shows this technique isn't merely accelerating convergence. Please add this to the paper appendix.

Negatives:
1. Missing comparisons to stronger baselines like DinoV2. This would admittedly require training on much larger pretraining datasets.

Reviewers:
The reviewers are positive about this work and maintain the positive ratings after the author rebuttal.

Decision:
Given the reviewer recommendations and results of this paper, it should be accepted at NeurIPS.
Further, given that it shows strong empirical fixes to I-JEPA this work should be made a spotlight.
However, since the proposed improvements are limited to a single method (rather than all non-contrastive methods like BYOL, DINO), the AC does not recommend it to be an oral.